Transcriptome atlas of Phalaenopsis equestris

Klepikova Anna V. annklepikova@gmail.com 1
Kasianov Artem S. 1
Ezhova Margarita A. 1 2
Penin Aleksey A. 1 3
Logacheva Maria D. 1 2 3
1 Laboratory of Plant Genomics, Institute for Information Transmission Problems of the Russian Academy of Sciences , Moscow , Russia
2 Skolkovo Institute of Science and Technology , Moscow , Russia
3 Lomonosov Moscow State University , Moscow , Russia
Balao Francisco
Electronic publication date: 2021 Dec 10
Publication date: 2021
Volume: 9
Electronic Location ID: e12600
Received 2020 Nov 26; Accepted 2021 Nov 15
Copyright: ©2021 Klepikova et al.
Copyright year: 2021
Copyright holder: Klepikova et al.
License: This is an open access article distributed under the terms of the Creative Commons Attribution License, which permits unrestricted use, distribution, reproduction and adaptation in any medium and for any purpose provided that it is properly attributed. For attribution, the original author(s), title, publication source (PeerJ) and either DOI or URL of the article must be cited.
License URL: https://creativecommons.org/licenses/by/4.0/

Keywords: Orchidaceae, Phalaenopsis equestris, Transcriptome map, RNA-seq, Database, Orphan genes

Funding: RFBR No. 18-29-13017 The reported study was funded by RFBR according to the research project No. 18-29-13017. The funders had no role in study design, data collection and analysis, decision to publish, or preparation of the manuscript.

==============================
The vast diversity of Orchidaceae together with sophisticated adaptations to pollinators and other unique features make this family an attractive model for evolutionary and functional studies. The sequenced genome of Phalaenopsis equestris facilitates Orchidaceae research. Here, we present an RNA-seq-based transcriptome map of P. equestris that covers 19 organs of the plant, including leaves, roots, floral organs and the shoot apical meristem. We demonstrated the high quality of the data and showed the similarity of the P. equestris transcriptome map with the gene expression atlases of other plants. The transcriptome map can be easily accessed through our database Transcriptome Variation Analysis (TraVA) for visualizing gene expression profiles. As an example of the application, we analyzed the expression of Phalaenopsis “orphan” genes–those that do not have recognizable similarity with the genes of other plants. We found that approximately half of these genes were not expressed; the ones that were expressed were predominantly expressed in reproductive structures.

Introduction

The enormous diversity of orchids traditionally attracts the attention of plant biologists. Orchidaceae comprises approximately 25 thousand species, which makes it the largest plant taxon (Cai et al., 2015). The diversification of orchids has evolved along with complex pollinator-adapted flower structure (Cozzolino & Widmer, 2005), CAM photosynthesis and epiphytism (Silvera et al., 2009). Orchids are highly valuable ornamental plants; among them, Phalaenopsis species are one of the most widely grown and sold (Griesbach, 2002). There is a need to develop tools for the identification of species and cultivars and for marker-assisted breeding.

A genome assembly of Phalaenopsis equestris (horse phalaenopsis) (Cai et al., 2015) and its improvement with long reads (Zhang et al., 2017) provided novel opportunities for evolutionary and functional studies of Orchidaceae. Genome assembly was used for functional studies of transcription factor families (Lin et al., 2016; Valoroso et al., 2019), somatic embryogenesis (Chen et al., 2019), and retrotransposon insertions (Hsu et al., 2019), as well as for evolutionary studies of ancient polyploidy (Barrett et al., 2019). However, transcriptome resources of P. equestris remain limited even though de novo transcriptome assembly was performed based on RNA sequencing of 11 organs (Niu et al., 2016). Transcriptome data are essential for many research directions. In particular, they provide insight into gene function in multigene families (see, e.g., Su et al., 2013; Kuo et al., 2021). Additionally, if available for different species or varieties, these data help to characterize genetic diversity in coding regions (see, e.g., Tsai et al., 2015).

In our study, we present a transcriptome map of P. equestris consisting of 19 samples in two biological replicates. The ornamental orchid P. equestris comprises three varieties and numerous hybrids of various flower colors and sizes (Hsu & Chen, 2016). To create a transcriptome atlas, we chose P. equestris var. cyanochilus because clonal plants are available for this cultivar, which helps to reduce interindividual variability (Fig. 1A). High-quality RNA of orchid organs and tissues was sequenced using Illumina technology, resulting in 1,687 M reads. We compared the expression characteristics of the P. equestris transcriptome map with gene expression atlases of other plants to provide evidence of the reliability of our data. Transcriptome map of P. equestris can be applied in a great variety of functional studies.

Figure 1 (A) The general view of P. equestris var. cyanochilus; (B) hierarchical clustering tree of transcriptome map samples; (C) the distribution of genes by the number of samples where the gene is expressed.

Only expressed genes with 5 or more normalized read counts in each biological replicate were considered; (D) the distribution of the Shannon entropy of P. equestris genes.

Materials & Methods

Growth conditions

Plants of P. equestris var. cyanochilus (the name of the variety means “blue lip”, var. blue, according to the provider, orchidee.su) were grown in a climate chamber under a 16 h light/8 h dark cycle at 22 °C and 50–60% relative humidity. Samples were collected in two biological replicates; each replicate consisted of at least seven plants. Sample collection was performed within two hours (Zeitgeber time ZT8-10) to reduce the influence of the circadian cycle.

RNA extraction, library preparation and sequencing

RNA was extracted using the RNeasy mini kit (Qiagen, The Netherlands) following the manufacturer’s protocol. To ensure a high quality of Phalaenopsis samples, RNA was analyzed using capillary electrophoresis on an Agilent Bioanalyzer 2100. cDNA libraries for Illumina sequencing were constructed using the NEBNext Ultra II RNA Library Prep Kit for Illumina (New England BioLabs, MA, USA) following the manufacturer’s protocol in 0.5 of the recommended volume (due to low RNA quantity in samples such as the shoot apical meristem). cDNA libraries were sequenced with HiSeq4000 and NextSeq500 (Illumina, CA, USA) instruments (50 bp and 75 bp single read runs).

Read mapping

Read trimming was performed using Trimmomatic version 0.36 (Bolger, Lohse & Usadel, 2014) in single read mode with the parameters “ILLUMINACLIP:common.adapters.file: 2:30:10 LEADING:20 TRAILING:20 SLIDINGWINDOW:4:15 MINLEN:30”. For read mapping, genome assembly and annotation of P. equestris from the PLAZA database (version 4.5, https://bioinformatics.psb.ugent.be/plaza/versions/plaza_v4_5_monocots/organism/view/Phalaenopsis+equestris) was used. Trimmed reads mapped on the genome assembly using Spliced Transcripts Alignment to a Reference (STAR) version 2.4.2 (Dobin et al., 2013) in the “GeneCounts” mode and parameters “–sjdbOverhang 59 –sjdbGTFfeatureExon exon –sjdbGTFtagExonParentTranscript gene_id” to obtain counts of uniquely mapped reads on each gene.

Expression characteristics of the transcriptome map

Gene read counts obtained with STAR were normalized to library size using size factors, as described in (Anders & Huber, 2010). A threshold of five or higher normalized read counts in each biological replicate was used to define expressed genes.

To describe the gene expression pattern, Shannon entropy values were calculated for expressed in at least one sample genes (Schug et al., 2005). To avoid overrepresentation of certain plant organs, the samples were grouped using distances on a clustering tree: gene expression levels were averaged if samples had distance (1 - Pearson r2) less than 0.1. Sample groups are listed in Table S1.

Differential expression analysis

Differential expression between each pair of samples was analyzed using the R packages “DESeq2” (Love, Huber & Anders, 2014), “edgeR” (Robinson, McCarthy & Smyth, 2010), and “baySeq” (Hardcastle & Kelly, 2010). The thresholds “FDR-corrected p-value <0.05” and “fold change ≥ 2” were used to consider a gene as differentially expressed.

Data availability

The raw RNA-seq data of the transcriptome map were deposited in the NCBI Sequence Read Archive (SRA) under BioProject accession PRJNA667255. The TraVA database can be accessed at http://travadb.org/browse/Species=Phalaenopsis_equestris/.

Results and Discussion

Transcriptome map construction

We collected 31 samples covering main plant organs and developmental stages, such as roots, young and mature leaves, floral organs, flower buds, and meristems. Each sample was collected in two biological replicates, and each replicate was pooled from at least seven plants. Sample RNA was sequenced on an Illumina platform, resulting in 29 M–65 M raw single reads (38 M median) for each sample (for sequencing statistics, see Table S2). After removing low-quality reads and adapter sequenced 98.7–99.8% of reads remained (Table S2).

Reads were mapped to the reference genome of P. equestris (Cai et al., 2015) with only one match allowed (unique mapping); 9.2–89.6% of high-quality reads were successfully mapped (Table S2). Twelve samples showed an extremely low mapping percentage; unmapped reads were identified as sequences belonging to Cymbidium mosaic virus (GenBank accession MK816927), which is known to persist in the majority of the P. equestris population and affect mainly mature and senescent tissues (Koh, Lu & Chan, 2014). As the library size of infected samples was insufficient and could distort the conclusions, we excluded samples with a percentage of mapped reads lower than 35% in at least one biological replicate. The remaining samples had 37.3–89.6% uniquely mapped reads with a median of 81.6%.

Thus, we constructed a transcriptome map of P. equestris covering 19 organs and parts of the plant. Floral organs (anthers, labellum, inner and outer tepals), leaves at different developmental stages, axes (inflorescence and pedicel), shoot apical and inflorescence meristems, and root parts were taken into the analysis (for a detailed description of the samples, see Table S3). The biological replicates showed high consistency (median Pearson r2 = 0.99, Table S4).

Clustering of samples generally reflects the plant body plan and groups organs with similar morphology and physiology (Klepikova & Penin, 2019). Hierarchical clustering of P. equestris samples showed the same pattern (Fig. 1B). Sample clusters were formed by floral organs, leaf parts, meristems and young leaves, inflorescence internode and root; young and mature anthers were the most distant from the other samples, similar to A. thaliana, rice, and maize (Nobuta et al., 2007; Wang et al., 2010; Stelpflug et al., 2016; Klepikova et al., 2016). The distances between samples on the clustering tree were closer than those in the other species we observed (Klepikova et al., 2016; Penin et al., 2019), which can be explained by the lack of older tissues in the P. equestris transcriptome atlas.

We compared our samples with publicly available P. equestris transcriptomes (Table S5). In general, the clustering of samples was consistent (Fig. S1), although leaves and columns from BioProject PRJNA288388 (Niu et al., 2016) clustered outside the other samples.

Expression characteristics of P. equestris

The Phalaenopsis genome annotation (PLAZA database, version 4.5) includes 29 431 protein-coding genes. Among them, 14 174 (48%) genes were expressed in all samples (using five reads in each biological replicate as a threshold), while transcripts of 21 671 (74%) genes were found in at least one sample. These values are in the range of typically expressed gene numbers across plant transcriptome maps (Klepikova & Penin, 2019). As in other species, samples demonstrated similarity in the number of expressed genes, which varied from 15 612 (53%) in the shoot apical meristem to 18 947 (64%) in ovules before pollination (Table S6).

Expression patterns of P. equestris genes

The study of gene expression patterns can shed light on the biological function of the gene and place it among essential for a plant existence ubiquitously expressed genes or precise regulators of tissue features –sample-specific expressed genes. We used two approaches to define gene expression patterns. Counting the number of samples where a gene is expressed is the simplest method to characterize expression pattern width, as was shown for Nicotiana tabacum (Edwards et al., 2010) and Vigna unguiculata (Yao et al., 2016). Here, the majority of genes (16,486) were expressed in 17 or more samples; the second peak (1,896 genes) of the distribution was formed by genes expressed in three or fewer samples (Fig. 1C). Most tissue-specific genes were found in anthers (56% of tissue-specific genes), roots (11%), and meristems (both shoot apical and inflorescence meristems, 8%). A high number of anther-associated genes have been found in A. thaliana (Klepikova et al., 2016) and are expected for P. equestris, as young and mature anthers are the most distant samples on the clustering tree (Fig. 1B).

While useful, this first approach depends on an arbitrary threshold that separates expressed and nonexpressed genes and does not take into account the variation in the expression level between samples. To overcome this issue, we used Shannon entropy as a measure of expression pattern width: low entropy values correspond to tissue-specific genes, while high values mark ubiquitously expressed genes (Schug et al., 2005). The distribution of Shannon entropy in P. equestris was significantly skewed to the right, revealing a large proportion of widely expressed genes (Fig. 1D), similar to A. thaliana, Solanum lycopersicum and Zea mays (Sekhon et al., 2013; Klepikova et al., 2016; Penin et al., 2019).

Using a Shannon entropy value lower than 0.25, we identified 521 tissue-specific genes. As was found under direct counting, the majority of genes were associated with anthers or roots (Fig. 2). According to GO enrichment, genes uniquely expressed in the mature anthers were involved in cell wall organization, biogenesis and modification and had pectin esterase and enzyme inhibitor activity (Table S7). Young anthers were characterized by genes encoding products with amine and amino acid binding activity (Table S8). Root-specific genes (expressed in the sample “Root without apex”) were described by the terms “response to chemical stimulus”, “response to oxidative stress”, “oxidation–reduction”, and “heme binding” (Table S9).

Figure 2 The heatmap of tissue-specific genes.

Expression levels of each gene in each sample were normalized on its maximal expression level.

To identify genes that were uniformly expressed across tissues, we selected genes with a Shannon entropy 3.55 or higher and calculated the coefficient of variance (CV) as a measure of expression stability. For 899 out of 1,340 genes, the CV was less than 0.25, indicating uniform expression in all samples and biological replicates. Stable genes had GO enrichment in terms associated with vesicles, membranes, RNA processing and localization. The list of GO categories strongly overlapped with the enrichment of uniformly expressed genes in A. thaliana, indicating interspecies universality of basic biological processes (Table S10).

P. equestris transcriptome variation database

We aimed to make our transcriptome data easily accessible and ready to use, so we uploaded P. equestris transcriptomes into our database Transcriptome Variation Analysis (TraVA, http://travadb.org/browse/Species=Phalaenopsis_equestris/). Gene IDs which are used in TraVA database (e.g., PEQU_41727) match the IDs in PLAZA genome assembly (see Materials and Methods). The TraVA interface demonstrates a color chart of gene expression profiles in a single- or multiple-gene view. A user can choose to show or hide expression values in a chart and choose between several types of read count normalization (Fig. 3).

Figure 3 Database view.

Expression profiles of P. equestris genes PEQU_39433, PEQU_02900, PEQU_33696..

We included the results of differential expression analysis in the TraVA database. In single-gene mode, the user can select one of three tools (DESeq2, edgeR or baySeq) and a sample against which other samples will be tested for the presence of differential expression.

Application of the TraVA database to the characterization of orchid genes

The graphical interface of TraVA facilitates gene expression pattern analysis and comparison and can be widely used in P. equestris functional studies. Orchids are a large and highly diverse plant family whose species are adapted to a number of ecological niches (typical terrestrial plants, epiphytes, nonphotosynthetic plants). These adaptations are reflected in their genome –for example, in P. equestris, which has a sophisticatedly differentiated perianth, the number of AP3 orthologs is higher than that of Apostasia shenzhenica, the basal orchid species with an undifferentiated perianth (Zhang et al., 2017). Conversely, P. equestris, which is an epiphyte and does not develop typical terrestrial roots, lacks AGL12 and several genes of the ANR1 clade, in contrast to A. shenzhenica (Zhang et al., 2017). This stresses the importance of the study of lineage-specific genes and gene families. For the genes that do not have orthologs in model species, the analysis of their expression profiles is the first step towards functional characterization.

We identified 181 (160 after filtering out the proteins that had Xs across more than 50% of its length) P. equestris proteins that did not have significant similarity to any Arabidopsis protein (e-value cutoff = 10). Of them, 118 share similarity with the proteins of A. shenzhenica and are thus presumably orchid-specific, while 42 have no hits and thus emerged after the divergence of Apostasioideae and Epidendroideae. The survey of the expression profiles showed that 93 of them were not expressed in any of the samples in the map (Fig. S2). Among those that are expressed, most are expressed at very low levels. Higher expression levels were associated with reproductive structures, particularly anthers (Fig. 4).

Figure 4 The heatmap of P. equestris-specific genes.

Expression levels of each gene in each sample were normalized on its maximal expression level for the color key. The numbers on the figure represent normalized gene read count averaged over biological replicates.

Among vegetative structures, the most distinct is the root (root apex), where three genes—PEQU_39433, PEQU_02900, and PEQU_33696—have the highest expression levels. Phalaenopsis roots are unique (compared to most other plants, including A. shenzhenica, but not other epiphytic orchids) in many respects—in particular, they are photosynthetic and develop a special structure called velamen. Velamen is a tissue of epidermal origin that consists of several layers of dead cells that help to absorb water and protect photosynthetic tissues of the root from UV damage. Notably, PEQU_39433 and PEQU_33696 do not have homologs in A. shenzhenica. PEQU_02900 has marginal similarity (34%) with the A. shenzhenica protein encoded by the Ash001570 gene.

The topic of orphan genes—those that lack detectable homologs in other lineages—is widely discussed, particularly regarding plants (Arendsee, Li & Wurtele, 2014). While some orphan genes might represent artifacts of the annotation, others have a function (for example, A. thaliana orphan gene QQS, which acts in starch metabolism) (Li et al., 2009). The functional analysis of orphan genes, however, lags behind that of typical genes, as orphan genes are overlooked in annotations based on homology; they are also usually expressed at lower levels and in a narrower range of tissues (reviewed in Schlötterer, 2015). The study of expression levels and patterns of a potential orphan gene is a first step towards its characterization—the detectable level of expression is evidence that the ORF is indeed a gene, not an annotation artifact.

Notably, orphan genes in well-characterized animal genomes (e.g., those of Drosophila and primates) have expression patterns biased towards male reproductive structures (Begun et al., 2007; Xie et al., 2012). According to the “out-of-testis” hypothesis (Kaessmann, 2010), this phenomenon is mediated by the unique epigenetic state of chromatin during male gametogenesis. We observed the same bias in Phalaenopsis; the growing availability of plant transcriptome maps will enable us to determine if this phenomenon is universal for plants.

Conclusions

In this study, we present a transcriptome map of the orchid Phalaenopsis equestris covering 19 organs at various stages of development. We identified 521 tissue-specific genes, the majority of which were expressed in anthers, roots (11%), and meristems. The uniformly expressed genes were associated with similar processes as those in Arabidopsis thaliana, i.e., vesicles, membranes, RNA processing and localization. To improve the use of these data, we integrated a transcriptome map into our database TraVA and demonstrated its usability in the study of P. equestris orphan genes. We expect that this resource will help to further investigate the genetic basis of unusual traits typical for this species and/or other orchids (CAM photosynthesis, unique floral structure, the development of aerial roots). Moreover, since Phalaenopsis equestris is a monocot and there is a bias towards grasses in large-scale transcriptome projects (see Klepikova & Penin, 2019), the characterization of the nongrass monocot transcriptome will be helpful not only for partial genetics of this species but also for large-scale comparative transcriptomics of flowering plants.

Supplemental Information

Supplemental Information 1 Supplementary Tables

See content on the first sheet of the file.

Click here for additional data file.

Supplemental Information 2 Hierarchical clustering tree of transcriptome map samples

Clustering of biological replicates.

Click here for additional data file.

Supplemental Information 3 The heatmap of all P.equestris-specific genes

Expression levels of each gene in each sample were normalized on its maximal expression level for the color key.

Click here for additional data file.

Additional Information and Declarations

Competing Interests

Author Contributions

Data Availability

The authors declare there are no competing interests.

Anna V. Klepikova analyzed the data, prepared figures and/or tables, authored or reviewed drafts of the paper, and approved the final draft.

Artem Sergeevich Kasianov analyzed the data, prepared figures and/or tables, and approved the final draft.

Margarita A. Ezhova performed the experiments, prepared figures and/or tables, and approved the final draft.

Aleksey A. Penin conceived and designed the experiments, performed the experiments, prepared figures and/or tables, and approved the final draft.

Maria D. Logacheva analyzed the data, authored or reviewed drafts of the paper, and approved the final draft.

The following information was supplied regarding data availability:

The RNA-seq raw data of transcriptome map are available at NCBI Sequence Read Archive (SRA): PRJNA667255.

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
