# Peer review of "Transcriptome atlas of Phalaenopsis equestris"

_PeerJ, doi:10.7717/peerj.12600_

## Round 0.1 · original submission · Major Revisions

Your paper has been reviewed by three referees and all agree that it will be worth publishing after revision (see reviews below).

Referees 1 and 3 pointed out the need to improve the introduction and conclusions sections. A native English speaker should proofread the manuscript.

Referee 2 suggests selecting several genes to verify whether the selected genes had a tissue-specific expression by qPCR.While I think this is a non-mandatory suggestion, I strongly suggest performing a Differential Expression analysis (DEseq, EdgeR, etc) to check DE genes on the different tissues. Authors could perform a Principal Component Analysis based on expression levels as a complementary analysis.

In your revision, please make sure to address all of the points raised by both reviewers on the current ms.

I look forward to receiving your revision in due course.

Sincerely,

Francisco Balao

·

Basic reporting

Orchids are one of the most beautiful and fascinating plant species. The high differences among orchids fascinated a large part of the scientific community during the centuries. Until now, the molecular pathways regarding orchid development are arduous to study for the absence of mutant of this species. The genome and transcriptome sequencing are two pivotal resources to understand which genes are involved in orchid differentiation. The data obtained with this study will be useful to all the scientific community that focus their analysis on orchids to understand which genes are involved in some unknown molecular pathways. The TraVa database will be helpful support for future orchid studies.
I suggest improving the introduction emphasizing the importance of this transcriptome data and why they will be important for the scientific community. It is necessary, in my opinion, to highlight why the transcriptome data is a relevant resource adding some work example where the use of transcriptome data was decisive to understand the role and evolution of some genes in orchid species.
The manuscript is clear and well written. I have listed some correction in the text in the attached file.
The figures are precise. I suggest, if it is possible, to change the interval colour to better highlight the differences of expression among genes, especially in Fig. 2.

Experimental design

No comment

Validity of the findings

I suggest adding to the conclusion the importance of this transcriptome sequencing, the TRaVa database and the orphan genes investigated. I recommend emphasizing on the importance of the data obtained and the use in future of these data.

Additional comments

L.26: Remove “a” between “about” and “half”
L.47: Add “the” before “reliability
L. 79: Substitute “In order to” with “To”
L. 102: Add “an” between “showed” and “extremely”
L. 112: Add “the” before “analysis”
L. 112: Add “the” before “detailed”
L. 129-130: I believe that there is a problem in this part “when transcripts of 21 671 (74%) genes were found in at least one sample.” Probably “when” needs to be changed in “while”. If there isn’t a problem, I suggest changing this phrase because I believe it is unclear.
L. 138-140: “A number of samples where gene is expressed is the simplest method to characterize expression pattern width, as was shown for Nicotiana tabacum (Edwards et al., 2010) or Vigna unguiculata (Yao et al., 2016)” it should be reformulated in “Investigate some samples, where a gene is expressed, is the simplest methods to characterize expression pattern width, as was shown for Nicotiana tabacum (Edwards et al., 2010) or Vigna unguiculata (Yao et al., 2016)”.
L. 142: Replace “less” with “fewer”
L. 146: Add “the” between “on” and “clustering”
L. 147: Replace “less” with “fewer”
L. 155: Add “the” between “in” and “case”
L.156: Add a comma after “count”
L. 162: Replace “oxidation reduction” with “oxidation-reduction”
L. 178: Add “the” before “Graphical”
L. 180: Substitute “number of” with “several”
L. 183: Add “the” between “with” and “undifferentiated”
L. 210: Add commas before and after “however”
L. 214: Remove “an” between “is” and “evidence”
L. 216: Add a comma after “Notably”
L. 226: Remove “the” before “similar”
L. 227: Substitute “In order to” with “To”

·

Basic reporting

no comment

Experimental design

no comment

Validity of the findings

no comment

Additional comments

I hope that the author can select several genes to verify whether the selected genes had a tissue-specific expression by qPCR or RT-PCR

·

Basic reporting

Some references should be added, I mentioned in the "General comments for the author".

Experimental design

No comment.

Validity of the findings

No comment.

Additional comments

Line 36: In the genome paper of Apostasia shenzhenica (Zhang et al., 2017),the author provided the new assembly of Phalaenopsis equestris under original NCBI accession. And this paper should be cited.
Line 90-93: I prefer to move this part to the Introduction. Although transcriptome and genome data were available for P. equestris, transcriptomes provided by author have more samplings and were from a variety named P. equestris var. blue, which helps to enrich the genetic resource for P. equestris. Thus, author should write two or three sentences to illustrate the ornamental value. Also, it will be better to provide the picture of P. equestris var. blue in the supplementary part since the potential utility on the molecular breeding.
Line 117-118: It will be better to change the word rather than "outgroup" since the cluster result is not a rooted tree. Some paper used this expression, but it is easy to make misleading.
Line 119-122: I can’t understand why the expression pattern would be similar for the lack of older tissues? Please illustrate the reason clearer.
Line 125: the same issue about “outgroup”. Use other statement likes “most divergent”or “form the sister cluster…”.
Line 185: "Apostasia shenzhenica", no "schenzhenica". And this paragraph discussed many about MADS box study in orchids, but without any citation.

---

## Round 0.2 · Minor Revisions

The authors have carefully addressed the reviewers' comments.

However, authors need to build connections to contigs and assembled data in a more open and clear manner. We do require assembled sequences to be deposited, as well as short reads (https://peerj.com/about/policies-and-procedures/#data-materials-sharing). This information will improve the TraVA database usability (currently, the gene ids are unknown).

Furthermore, I was able to access the results of differential expression analysis in the TraVA database.

After these recommended, minor revisions, the manuscript will be acceptable for publication.

·

Basic reporting

No comment

Experimental design

No comment

Validity of the findings

No comment

Additional comments

No comment

·

Basic reporting

no comment

Experimental design

no comment

Validity of the findings

no comment

Additional comments

no comment

·

Basic reporting

All pass. I can't judge well about the English expression but I found the sentences can be read easily and didn't make misunderstood. The author cited the related papers as I mentioned in last reviewing.

Experimental design

Pass. The author revised the manuscript according to my comments.

Validity of the findings

Pass. The author revised the manuscript according to my comments.

Additional comments

Pass. The author revised the manuscript according to my comments.

---

## Round 0.3 · accepted · Accept

Thank you for submitting the revised version of your manuscript. I recommend that this study could be accepted in PeerJ.